# Novel Dental Poly (Methyl Methacrylate) Containing Phytoncide for Antifungal Effect and Inhibition of Oral Multispecies Biofilm

**DOI:** 10.3390/ma13020371

**Published:** 2020-01-13

**Authors:** Myung-Jin Lee, Min-Ji Kim, Sang-Hwan Oh, Jae-Sung Kwon

**Affiliations:** 1Department and Research Institute of Dental Biomaterials and Bioengineering, Yonsei University College of Dentistry, Seoul 03722, Korea; lmj239@yuhs.ac (M.-J.L.); dmmj@yuhs.ac (M.-J.K.); 2BK21 PLUS Project, Yonsei University College of Dentistry, Seoul 03722, Korea; 3Department of Dental Hygiene, College of Medical Science, Konyang University, Daejeon 35365, Korea; dentsh@hanmail.net

**Keywords:** poly (methyl methacrylate), phytoncide, antifungal, dental polymer, oral biofilm

## Abstract

Despite the many advantages of poly (methyl methacrylate) (PMMA) as a dental polymer, its antifungal and antibacterial effects remain limited. Here, phytoncide was incorporated into PMMA to inhibit fungal and biofilm accumulation without impairing the basic and biological properties of PMMA. A variable amount of phytoncide (0 wt % to 5 wt %) was incorporated into PMMA, and the basic material properties of microhardness, flexural strength and gloss were evaluated. In addition, cell viability was confirmed by MTT assay. This MTT assay measures cell viability via metabolic activity, and the color intensity of the formazan correlates viable cells. The fungal adhesion and viability on the PMMA surfaces were evaluated using *Candida albicans* (a pathogenic yeast). Finally, the thickness of saliva-derived biofilm was estimated. The flexural strength of PMMA decreased with increasing phytoncide contents, whereas there were no significant differences in the microhardness and gloss (*p* > 0.05) and the cell viability (*p* > 0.05) between the control and the phytoncide-incorporated PMMA samples. The amounts of adherent *Candida albicans* colony-forming unit (CFU) counts, and saliva-derived biofilm thickness were significantly lower in the phytoncide-incorporated PMMA compared to the control (*p* < 0.05). Hence, it was concluded that the incorporation of appropriate amounts of phytoncide in PMMA demonstrated antifungal effects while maintaining the properties, which could be a possible use in dentistry application such as denture base resin.

## 1. Introduction

Poly (methyl methacrylate) (PMMA) has been used as polymer material in dentistry since 1937, to fabricate the denture base due to the various advantages, such as good mechanical properties, low cost and low toxicity [1]. With an aging population, the number of denture users are expected to increase gradually, which would consequently increase the expectations from such polymer [2]. The prerequisite for polymer used as the denture base material is to have an excellent biocompatibility and high wear resistance, in addition to stable physical properties [3]. In particular, mechanical property is an important factor, as the denture base should be resistant to fracture and mastication force [4]. Also, the biological property of denture base resin is a crucial factor of a removable denture, as the acrylic resin such as PMMA is not only prone to allergic reactions in the oral mucosa, but also the residual monomer has potential cytotoxicity [5].

Even though PMMA is widely used as a denture base resin, it also has many inherent limitations. One of the main drawbacks is a poor antifungal and antibacterial property, which results in microorganisms being able to attach and accumulate into the biofilms [6,7]. Denture surfaces act as a reservoir for microorganisms, and may lead to diseases such as denture stomatitis [6]. Accordingly, the antifungal properties of denture base resins play an important role in the reduction of denture stomatitis [7]. As the PMM- based denture surface is hydrophobic, *Candida albicans*, which has a key role in the pathogenesis of denture stomatitis, would adhere more easily on the denture surface [8]. Moreover, scratches of the denture surface could accelerate microbial accumulation and cause denture stomatitis. Efforts have been underway to develop antifungal denture base resin to prevent biofilm colonization [9].

Previously, PMMA containing silver nanoparticles had a successful antifungal effect, and no cytotoxicity was found. However, these materials have the drawback of low mechanical properties [10]. Also, Thymoquinone (TQ) extracted from the Nigella sativa seed, was shown to inhibit the growth of *C. albicans* when incorporated with PMMA [11]. Still, the use of TQ has been limited, as the component was shown to be cytotoxic [12]. Clinically, a good balance of mechanical property and biocompatibility of denture base resin forms the backbone of successful denture base resin. Hence, it is necessary to improve PMMA with an effective antifungal effect and proper mechanical property.

Recently, natural substances have been studied because of their low residual toxicity and low side effects. Many researchers have been actively studying compounds in dental material science, which contain natural extracts such as *Curcuma xanthorrhiza* (Javanese ginger) or green tea [13]. Among these, the antimicrobial activity of phytoncide has been widely studied in recent years. Phytoncide is the extracted component of many different trees, including *Chamaecyparis obtusa* (Japanese cypress) [14]. Phytoncide has an antibacterial and antifungal effect against microorganisms [15]. Previous studies have shown that phytoncides not only have antimicrobial effects against Gram-positive bacteria, Gram-negative bacteria and fungi, but also play a crucial role in increasing the bacterial susceptibility to antibiotics [14,16]. Despite these many studies, there have been no attempts to incorporate phytoncide to dental polymer materials such as PMMA.

Therefore, the aim of this study was to incorporate a phytoncide with PMMA that would improve the antifungal effect and prevent biofilm accumulation without impairing the basic and biological property of PMMA. The null hypotheses were; (1) there would be no difference in antifungal effect and biofilm accumulation between the control PMMA and PMMA incorporating phytoncide, (2) there would be no differences in basic material properties such as microhardness, flexural strength and gloss between the control PMMA and PMMA incorporating phytoncide, and (3) there would be no difference in cytotoxicity between the control PMMA and PMMA incorporating phytoncide.

## 2. Materials and Methods

### 2.1. Incorporation of Phytoncide into PMMA

A commercially available, auto-polymerizing acrylic resin system (ProBase Cold, Ivoclar vivadent, Schaan, Leichtenstein) was used. Commercially-available phytoncide liquid was purchased from Pyeonbaekcide Co. Ltd. (Kim Min-jae Hinoki Cypress, Pyeonbaekcide Co. Ltd., Seongbuk-gu, Seoul, Korea). The substance is a hinoki cypress distillate which has been prepared as a mixture of oil and distilled water. The phytoncide was mixed with the acrylic resin liquid monomer at various weight percentages; 0% (control), 1.25%, 2.5%, 3.75% and 5%, respectively (Table 1). Phytoncide liquid and monomer were mixed under sonication for 1 h and then blended by automatic stirring for 4 h. All specimens were prepared in a powder to liquid mass ratio of 3:2 [17]. After low-temperature polymerization (60 °C, 4.0 bar, 15 min, Air Press Unit, Sejong Dental, Daejeon, Korea), specimens were obtained with different shapes for each experiment and polished with silicone carbide (SiC) papers 800 up to 2000 grit.

### 2.2. Chemical Characterization of Phytoncide-Incorporated PMMA

The chemical characterization of the control, 1.25% PT, 2.5% PT, 3.75% PT and 5% PT samples were analyzed by Fourier-transform infrared (FTIR) spectroscopy (FT/IR-4700, JASCO, Easton, MD, USA). Before measurement, the samples were washed twice with ethanol to remove impurities.

### 2.3. Mechanical Properties

#### 2.3.1. Microhardness

The samples were placed in a Vickers hardness tester (Microhardness tester, Dmh-2; Matuzawa Seiki, Tokyo, Japan), and a 50 g weight was applied for 10 s at different points of the samples. The indentation was observed, and the Vickers hardness number (VHN) was measured to determine the surface hardness. Two sites were measured at random for each specimen, and total of three specimens were tested for each group (n = 3), and the mean value and standard deviation were obtained and compared.

#### 2.3.2. Flexural Strength

The mechanical properties were measured according to ISO 20795-1 [18]. Five groups of samples were fabricated, with dimensions of 64 × 10 (*b*) × 3.3 (*h*) mm^3^. A computer-controlled universal testing machine (Model 3366; Instron^®^, Norwood, MA, USA) was used to fracture the specimens in three-point flexure. The flexural strength was measured at a span length of 50 mm and crosshead speed of 5 mm/min. The flexural strength was calculated as
(1)$$\sigma \;=\frac{3\mathrm{Fl}}{2bh^{2}}$$
where F is the maximum load, l is the distance between the supports, *b* is the width and *h* is the height.

#### 2.3.3. Gloss

Surface gloss was measured using a Novo-curve gloss meter (Novo-Curve, Rhopoint Instrumentation, East Sussex, UK) in gloss units (GU). Each specimen was fabricated to a thickness of 2 mm and a diameter of 10 mm using a Teflon mold. The specimens were polished using silicon carbide paper in sequential order of 800, 1000 and 1200 grit sizes. Specimens were mounted on the plate for the measurements. Measurements were obtained from two specimens of each group, and five points were randomly measured for each specimen. The flat surface and small area were measured using a 60° geometry.

### 2.4. Biological Properties

The cytotoxicity was evaluated using the MTT (3-(4, 5-dimethylthiazol-2-yl)-2, 5-diphenyl tetrazolium bromide) assay using murine fibroblast of L929 (Korean Cell Line Bank, Seoul, Korea), in accordance with the International Standard, ISO 10993-5 [19]. Briefly, samples were extracted in culture medium for 72 h at 37 °C. L929 cells were seeded at a density of 1 × 10^5^ cells/mL into 96-well plates (SPL, Pochen-Si, Gyeonggi-Do, Korea) and cultured for 24 h.

Following removal of the culture medium and washing with Dulbecco’s phosphate-buffered saline (DPBS) (Gibco, Grand Island, NY, USA), 100 µL of extractions from samples and the negative control of the culture medium were placed into each well. Additionally, 100 µL of phenol was used as the positive control, which is known for its cytotoxicity. After 24 h, the culture medium and phenol were removed and replaced with 50 µL of MTT solution (Sigma-Aldrich, St. Louis, MO, USA) in phosphate-buffered saline (PBS). After 2 h, the MTT solution was discarded, and 100 µL of isopropanol (Sigma-Aldrich) was added to each well to solubilize the formazan product. The absorbance was then measured using an enzyme-linked immunosorbent assay (ELISA) reader (Epoch, BioTek, Winooski, VT, USA) at 570 nm. The cell viability was calculated as the percentage of the optical density measured for the negative control. For microscopic observation, the images of L929 cells exposed to sample extracts were observed using an EVOS FL microscope (Advanced Microscopy Group USA Ltd., Mill Creek, WA, USA) at 20× magnification. 

### 2.5. Antifungal and Oral Biofilm-Related Properties

#### 2.5.1. Colony Forming Units

The strain used in the experiment was *Candida albicans* (*C. albicans*, ATCC 10231). *C. albicans* was cultured in yeast mold (YM, Becton Dickinson and Co., Franklin Lakes, NJ, USA) broth at 37 °C for 24 h. After preparing the disk-shaped specimens, 1 mL of fungal suspension (1 × 10^8^ cells/mL) was placed onto each disk in a 24-well plate and incubated at 37 °C for 24 h, under a more than 95% humidified atmosphere. After incubation, the samples were gently washed twice with PBS to remove any non-adherent fungi. To evaluate the fungal colony-forming units (CFUs), the adherent fungi were harvested in 1 mL of YM by sonication (SH-2100; Saehan Ultrasonic, Seoul, Korea) for 5 min, where the procedures have been adapted from the previous study [20,21]. Next, 100 µL of this fungal suspension was spread onto a YM agar plate and incubated at 37 °C for 24 h. The total number of colonies was then counted.

#### 2.5.2. Fungal Viability

The viability of the adherent fungi was determined by staining using a LIVE/DEAD FungaLight Yeast Viability Kit (Molecular Probes, Eugene, OR, USA) according to the manufacturer’s protocols [20]. Equal volumes of syto 9 dye and propidium iodide from the kit were mixed thoroughly (where these two stain live and dead fungi, respectively). Subsequently, 3 µL of the mixture was added to 1 mL of the fungal suspension prepared as described above. After 15 min of incubation at room temperature in the dark, the stained samples were observed by confocal laser microscopy (CLSM, LSM880; Carl Zeiss, Thornwood, NY, USA). Live fungi appeared green, while dead fungi appeared red.

#### 2.5.3. Saliva-Derived Biofilm Thickness

For this experiment, the protocol from a previous study was adapted [21]. The saliva donors were six adults without dental caries or periodontal disease who had not consumed antibiotics in the past three months. The participants did not brush their teeth for 24 h and did not eat or drink for at least 2 h before donating saliva. Human saliva was collected from the donors and mixed in equal proportions. The mixed saliva was then diluted in sterile glycerol to a concentration of 30% and stored at –80 °C to be used as a biofilm model [22]. As human saliva is a very complex system, the McBain medium was used to simulate the saliva environment to obtain a stable environment for microbial growth [23]. The biofilm model was cultured in McBain medium supplemented with mucin (type II, porcine, gastric) (2.5 g/L), bacteriological peptone (2.0 g/L), tryptone (2.0 g/L), yeast extract (1.0 g/L), NaCl (0.35 g/L), KCl (0.2 g/L), CaCl2 (0.2 g/L), cysteine hydrochloride (0.1 g/L), hemin (0.001 g/L) and vitamin K1 (0.0002 g/L) at 37 °C for 24 h [24]. From the cultured medium, 1.5 mL of the bacterial solution was placed on the specimen. Following 8, 16 and 24 h of incubation, an additional 1.5 mL of the microbial solution was placed on the specimen after each period. Biofilms were allowed to grow for a total of 48 h. 

The specimens were stained with a live–dead bacterial viability kit (Molecular Probes, Eugene, OR, USA) via the same method described above for bacterial staining. The biofilm was then visualized at five randomly chosen positions using CLSM. Images of the axially stacked biofilm were acquired, and the thickness of each biofilm was measured using software (Zen, Carl Zeiss, Thornwood, NY, USA). The live and dead multispecies bacteria were quantified by ImageJ software (NIH, Bethesda, MD, USA).

### 2.6. Statistical Analysis

All statistical analyses were conducted using the SPSS 23 software program (IBM Corp., Armonk, NY, USA), and the level of significance was set at *p* < 0.05. The results among different groups were analyzed by one-way analysis of variance (ANOVA) followed by Tukey’s post hoc test.

## 3. Results

### 3.1. Chemical Characterization of Phytoncide-Incorporated PMMA

Figure 1. shows the chemical structures of control, 1.25% PT, 2.5% PT, 3.75% PT and 5% PT samples. Overall, the FTIR spectra appear similar because poly (methyl methacrylate) (PMMA) is the major component in the polymer matrix. In addition, the signature peaks of phytoncide were overlapped with the various peaks of MMA; MMA is colorless liquid which is a monomer produced for the PMMA. When polymerization occurs, MMA converts to PMMA. For example, the PO^4−^ peak at 1058 cm^−1^ and the C–O–C peak at 1060 cm^−1^. However, the absence of a carbon double bond peak at 1660 cm*^−^*^1^ indicates the successful conjugation of the phytoncide into the polymer matrix. 

### 3.2. Mechanical Properties

In terms of the microhardness and gloss, there were no significant differences among the different PMMA groups (*p* > 0.05), indicating that the incorporation of phytoncide did not affect the microhardness and gloss of the PMMA (Figure 2A and Figure 3B, bottom right). For the flexural strength, the results showed a decreasing trend with increasing amounts of phytoncide (*p* < 0.05; Figure 2B). Specifically, the 2.5% PT (flexural strength: 64.05 ± 5.62 MPa), 3.75% PT (56 ± 7.87 MPa) and 5% PT samples (47.15 ± 5.20 MPa) showed significantly lower values than those of the control samples (flexural strength: 83.25 ± 4.64 MPa). Additionally, only the 1.25% PT and 2.5% PT groups along with the control fulfilled the minimum requirements (60 MPa) specified in the ISO standard [18]. Finally, there were no significant differences in morphology between the control and all of the test samples as observed by the scanning electron microscopy (SEM) images (Figure 3B).

### 3.3. Biological Properties

All groups showed near 100% cell viability (Figure 4B). Microscopic observations showed that the positive control (PC) group containing phenol showed a round shape with cytoplasmic condensation and dendritic constriction (Figure 4A). Meanwhile, the groups other than PC showed a normal form of cells (Figure 4A).

### 3.4. Antifungal and Oral Biofilm-Related Properties

The CFU counts for the control PMMA were significantly higher than those for the other groups (Figure 5B). In addition, these findings were confirmed by fungal cell viability staining results, which showed much fewer live fungi on the surfaces of PMMA with phytoncide incorporated than those on the control PMMA (Figure 5A). Furthermore, the saliva-derived biofilm accumulation was consistent with the results obtained for *C. albicans*, indicating a lower abundance of multispecies bacteria on the surfaces of PMMA with the phytoncide incorporated, than that on the control PMMA (Figure 6). The incorporation of phytoncide significantly decreased the thickness of the biofilm on the surfaces of the PMMA (*p* < 0.05; Figure 6B).

## 4. Discussion

Although PMMA is widely used in denture base resin, it has limitations due to its lack of antimicrobial and antifungal activity [25,26]. The use of chemical antibiotics has led to a widespread understanding of the utility of natural substances with antimicrobial effects due to the severity of antibiotic-resistant bacteria and the toxicity of chemical components, and several attempts have been made for clinical use [27]. Among the explored compounds, phytoncides have been reported to show antimicrobial effects against various microorganisms such as bacteria and fungi [16], and they have been shown to be clinically effective against bad breath and periodontal disease, exhibiting antibacterial effects [15,28]. However, studies on the antimicrobial effect of phytoncide-incorporated PMMA are lacking. Therefore, in the present study, phytoncide was incorporated into PMMA to provide antifungal effects, and the fundamental and biocompatible properties of PMMA were evaluated.

In this study, a microhardness tester was used to determine the effects of phytoncide on the surface hardness. Hardness is related to wear resistance and is the most common mechanical property indicator for prosthetic materials [29]. The results of the statistical analyses have shown that the surface microhardness values between the experimental and control groups are not significantly different (*p* > 0.05; Figure 2A). This result implies that incorporating phytoncide does not have any effects on the surface hardness of PMMA. The mechanical properties of dentures play an important role in clinical performance [3]. For the flexural strength, the 1.25% PT and 2.5% PT groups along with the control fulfilled the minimum requirements (60 MPa) specified in the International Standard of ISO 20795-1 [18]. As the 3.75% PT and 5% PT groups showed low flexural strength, these experimental groups may be limited in terms of direct clinical application. When incorporating new materials into PMMA-based devices, the noticeable opacity and loss of gloss affect their clinical usefulness. Thus, gloss measurement is used to evaluate the esthetic appearance. In this study, the gloss results showed that there was no significant difference among the sample groups (*p* > 0.05; Figure 3B).

Most natural substances that have been used for various purposes have had their safety proven [13]. In addition, a wide range of the oral mucosa may be exposed to toxic or irritating compounds while using dentures [5]. Cell viability studies are designed to determine the biological response of mammalian cells in vitro using appropriate biological parameters [5,30]. To analyze the cell viability, the MTT assay was used, which is a well-established method for studying biocompatibility in terms of cytotoxicity endpoint [5,30]. For the MTT assay, the extraction procedure of the test materials for one day was followed by cell exposure to the extract for an additional day, as specified in the International Standard of ISO 10993-5 [30]. The results of the in vitro cytotoxicity test (Figure 4) showed that there were no significant differences between the experimental and control groups in terms of the cell viability of extracts (*p* > 0.05), where most of the experimental groups showed near 100% cell viability in comparison to the negative control. This would mean that the samples would be classified as noncytotoxic. It is well known that the incomplete polymerization of PMMA and the leaching of monomers would have a negative impact on the biocompatibility result in cytotoxicity [19]. This result implies that the phytoncide components do not have significant effects on the polymerization of the PMMA or the leaching of the monomer that would have effects on cell viability. This has been in fact confirmed by the chemical characterization of the specimen (Figure 1) which showed that there is a successful conjugation of the phytoncide into the polymer matrix.

The pathogenic microorganisms on dentures are associated with diseases such as denture stomatitis and aspiration pneumonia [31]. In particular, *C. albicans* is the key fungus frequently occurring in dentures [25]. The results confirmed a significant reduction in fungal attachment on the surfaces of phytoncide-incorporated PMMA compared to those of the control, as revealed by the CFU count. Additionally, the CLSM images and thickness of the formed multispecies biofilm of phytoncide-incorporated PMMA exhibit a similar trend with the CFU count (Figure 5). The results of this study confirmed a significant reduction in *C. albicans* adhesion on the surfaces of the phytoncide-incorporated PMMA compared to that of the control PMMA. Considering that removable devices are typically worn in a complex oral environment for a prolonged period, inhibition of biofilm formation to maintain a healthy oral environment is an effective way to prevent oral disease [26]. In the oral cavity, many microorganisms exist in the form of colonies, and approximately 500 kinds of microorganisms such as bacteria, viruses and fungi are known to exist in the oral cavity [8]. Thus, we used human saliva for the biofilm analysis because human saliva maintains the complexity and heterogeneity of dental plaque in vivo; therefore, it is an ideal material for the cultivation of biofilms in vitro [21]. The thickness of the formed multispecies biofilm of phytoncide-incorporated PMMA showed a significant reduction compared to that of the control PMMA (Figure 6).

In previous studies, it was supposed that phytoncide exerted antimicrobial effects by weakening the cell wall, causing bacterial self-degradation due to the destruction of cell membranes or affecting the respiratory metabolism of bacteria [28]. Thus, the released phytoncides from phenolic compounds may contribute to the inhibition of microbial activity [28]. In addition, most secondary metabolites produced by plants cause stress to various microorganisms. This stress affects microorganisms, and this phenomenon is recognized as a defense mechanism of plants. This phenomenon is called allelopathy, and is a factor in ecological processes [15]. Allelochemicals, the chemical substances involved in the allelopathy effect, are composed of essential components that can be obtained by extraction. The antimicrobial allelochemic organic compounds contained in these essential components are called phytoncides, and we assumed that phytoncide-incorporated PMMA exhibited antifungal effects through this mechanism [14,15]. Although the precise mechanism of phytoncides is unclear, these results show that such compounds might present effective antifungal properties. Further research is needed to elucidate the antimicrobial mechanisms and study the long-term effects during clinical application.

Based on the above results, the null hypothesis was partially rejected because there were significant differences in the basic material, biological and antifungal properties between the phytoncide-incorporated PMMA and the PMMA groups lacking phytoncide, while significant improvement in antifungal effect and oral biofilm reductions were observed with PMMA incorporated with phytoncide compared to the control PMMA. This study demonstrates one aspect of the combination of phytoncides and dental materials, and these basic data have the potential for use in the future development of PMMA possessing antifungal effects.

## 5. Conclusions

Within the limitations of this study, the results demonstrated that phytoncide-incorporated PMMA showed antifungal effects against *C. albicans*, with no deterioration of surface microhardness, flexural strength or gloss. Furthermore, the material was biocompatible in terms of cytotoxicity. Also, the addition of 2.5 wt % phytoncide was an optimal concentration about antifungal effect and the satisfying of proper physical properties. Hence, phytoncide-incorporated PMMA has the potential to minimize antifungal infection, and the material may be used as a dental polymer such as in denture base resin, providing that future additional studies are carried out.

## Figures and Tables

**Figure 1 materials-13-00371-f001:**
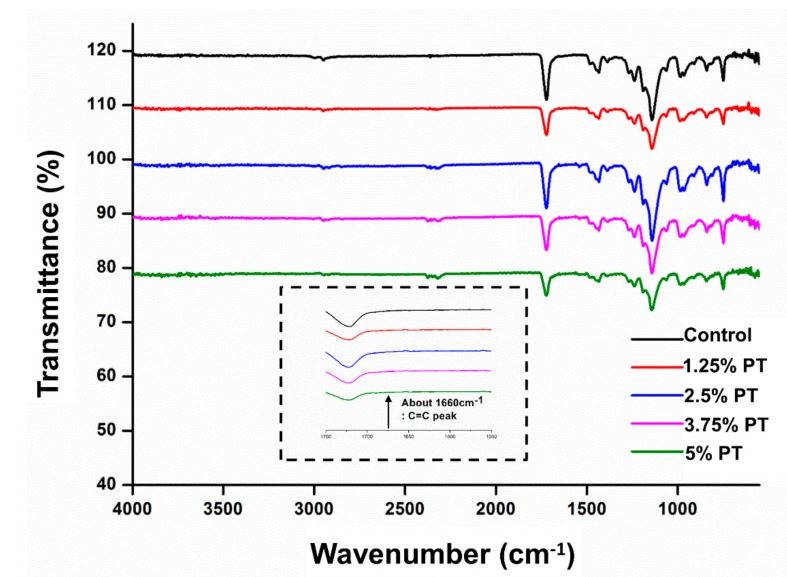
Chemical characterization of poly (methyl methacrylate) (PMMA)-based materials. Fourier transform infrared (FTIR) spectra of control, 1.25% PT, 2.5% PT, 3.75% PT and 5% PT.

**Figure 2 materials-13-00371-f002:**
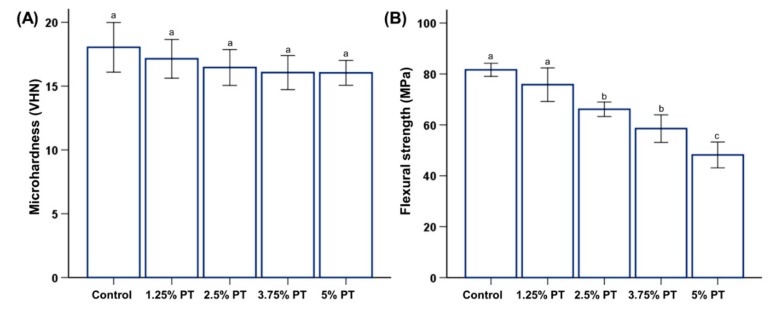
Comparison of microhardness (**A**) and flexural strength (**B**) among different groups of PMMA samples. Same lowercase alphabetical letters above the bar graph indicate there are no significant differences between the groups (e.g., there are no difference between group with ‘a’ and another group with ‘a’ above the bar) (*p* > 0.05). Difference in lowercase alphabetical letters above the bar graph indicate significant differences between the groups (e.g., there are differences between the group with ‘a’ and the group with ‘b’.) (*p* < 0.05).

**Figure 3 materials-13-00371-f003:**
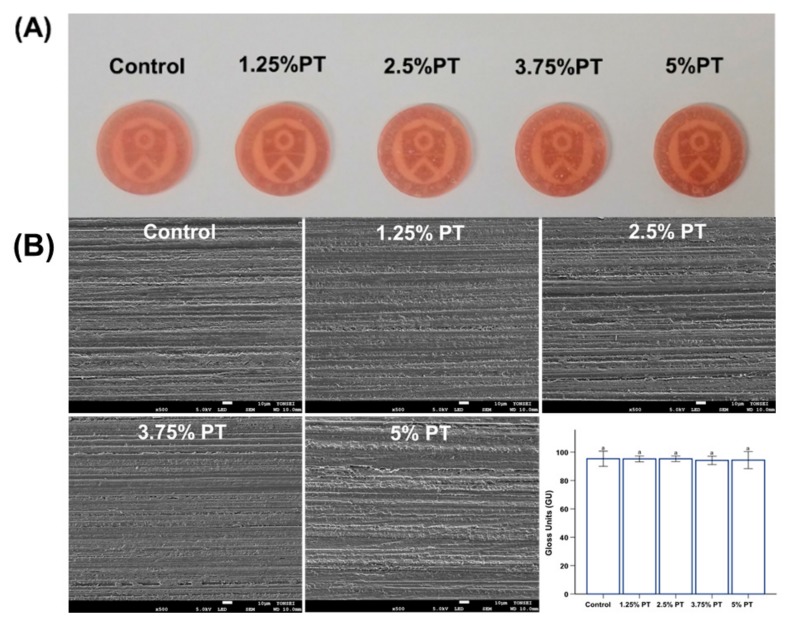
Representative images of PMMA samples; control, 1.25% PT, 2.5% PT, 3.75% PT and 5% PT (**A**), comparison of gloss values between different groups of PMMA samples (**B**). Same lowercase alphabetical letters above the bar graph indicate there are no significant differences between the groups (e.g., there are no differences between group with ‘a’ and another group with ‘a’ above the bar) (*p* > 0.05). Differences in lowercase alphabetical letters above the bar graph indicate significant differences between the groups (e.g., there are differences between the group with ‘a’ and the group with ‘b’.) (*p* < 0.05).

**Figure 4 materials-13-00371-f004:**
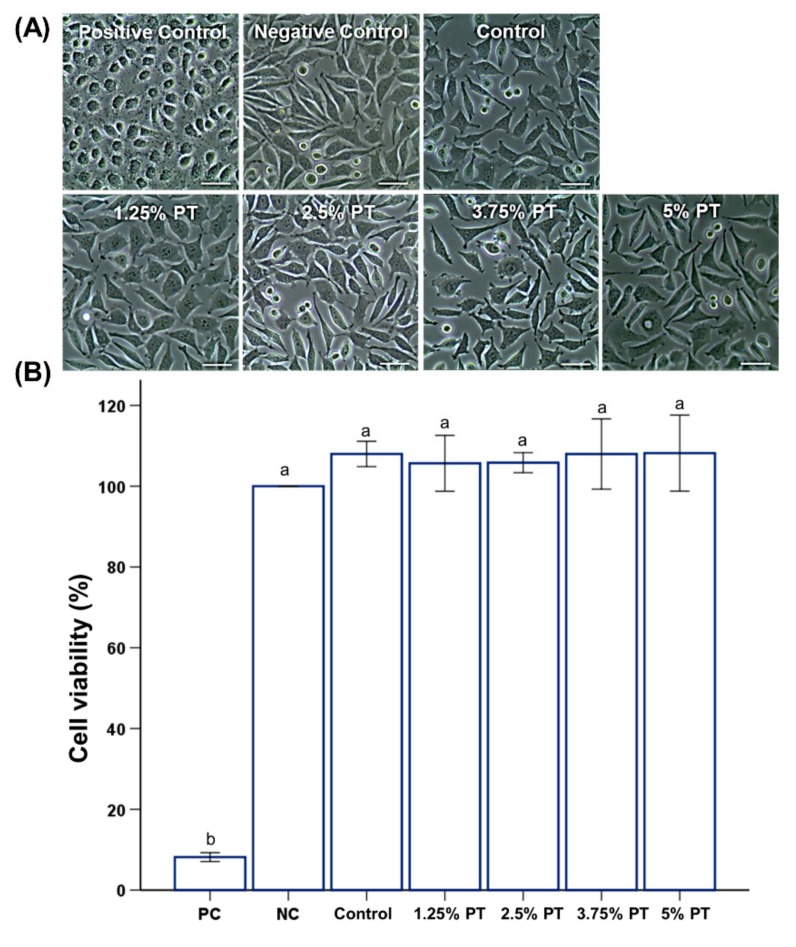
Representative microscopic images of L929 cells on the surfaces of PMMA samples; control, 1.25% PT, 2.5% PT, 3.75% PT and 5% PT at a magnification of 20× (**A**), comparison of cell viability among different groups of PMMA samples (**B**). Same lowercase alphabetical letters above the bar graph indicate there are no significant differences between the groups (e.g., there are no differences between group with ‘a’ and another group with ‘a’ above the bar) (*p* > 0.05). Differences in lowercase alphabetical letters above the bar graph indicate significant differences between the groups (e.g., there are differences between the group with ‘a’ and the group with ‘b’.) (*p* < 0.05).

**Figure 5 materials-13-00371-f005:**
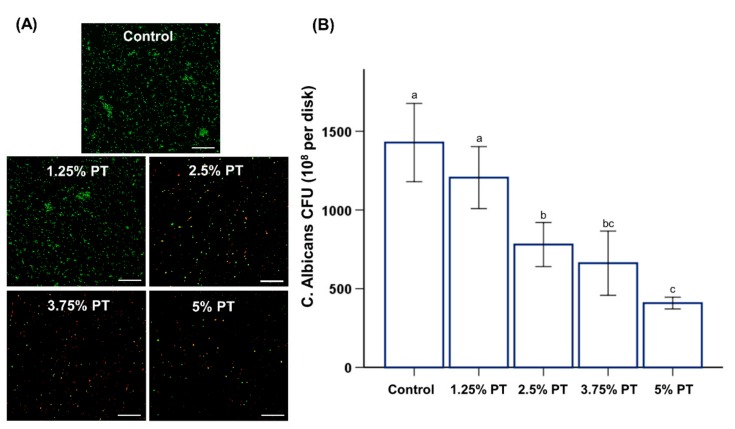
Representative live–dead staining images of *C. albicans* attached on the surfaces of PMMA samples; control, 1.25% PT, 2.5% PT, 3.75% PT and 5% PT at a magnification of 5000× (**A**), Colony-forming units (CFU) counts derived from fungi attached on the surfaces of PMMA samples (**B**). Same lowercase alphabetical letters above the bar graph indicate there are no significant differences between the groups (e.g., there are no differences between group with ‘a’ and another group with ‘a’ above the bar) (*p* > 0.05). Differences in lowercase alphabetical letters above the bar graph indicate significant differences between the groups (e.g., there are differences between the group with ‘a’ and the group with ‘b’.) (*p* < 0.05).

**Figure 6 materials-13-00371-f006:**
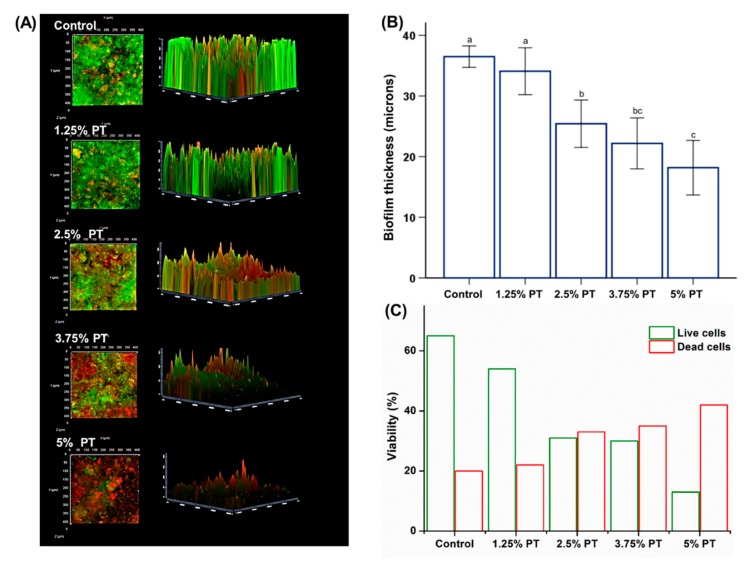
Representative saliva-derived biofilm accumulation images of *C. albicans* attached on the surfaces of PMMA samples; control, 1.25% PT, 2.5% PT, 3.75% PT and 5% PT at a magnification of 5000× (**A**), Biofilm thickness derived from fungi attached on the surfaces of PMMA samples (**B**), Live–Dead assay quantified as the live and dead cells in equivalent surface areas (**C**). Same lowercase alphabetical letters above the bar graph indicate that there are no significant differences between the groups (e.g., there are no differences between the group with ‘a’ and another group with ‘a’ above the bar) (*p* > 0.05). Differences in lowercase alphabetical letters above the bar graph indicate significant differences between the groups (e.g., there are differences between the group with ‘a’ and the group with ‘b’.) (*p* < 0.05).

**Table 1 materials-13-00371-t001:** Composition of materials in the control and experimental groups.

Groups	Group Code	PMMA (wt %)	Phytoncide (PT) (wt %)
1	Control	100	0
2	1.25% PT	98.75	1.25
3	2.5% PT	97.5	2.5
4	3.75% PT	96.25	3.75
5	5% PT	95.0	5

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
