# Peer review of "Novel Dental Poly (Methyl Methacrylate) Containing Phytoncide for Antifungal Effect and Inhibition of Oral Multispecies Biofilm"

_materials, 2020, doi:10.3390/ma13020371_

Round 1

Reviewer 1 Report

PMMA was shown to have over 100% viability on the L-929 cells in the PMMA and PMMA containing groups in the current study. Why PMMA did not show any cytotoxicity to the cells, which does not make sense. Why only biofilm thickness was calculated? The thickness cannot be equal to the antibiofilm effect.  The CFU test is only for planktonic bacteria while confocal can show live/dead percentage of biofilm. A result of the percentage of live/dead bacteria would be more promising showing the killing effect on biofilm.

Why L929 cells were used?

Why only two sites were measured on each sample for microhardness test? How did the authors determine this number?

Author Response

Reviewer #1

Q 1)

PMMA was shown to have over 100% viability on the L-929 cells in the PMMA and PMMA containing groups in the current study. Why PMMA did not show any cytotoxicity to the cells, which does not make sense.

A 1)

Thank you for your comments on the manuscript.

100% viability indicates that there was no cytotoxicity with PMMA. According to the previous study (reference below), cytotoxicity is caused by unpolymerized residual monomer, MMA. In the case of thermopolymerization as with the current materials considered, residual monomers are minimized. It is thought that there was no cytotoxicity due to the nature of polymerization.

We included this information in Discussion (text colored red in Line 288~304)

Q 2)

Why only biofilm thickness was calculated? The thickness cannot be equal to the antibiofilm effect.  The CFU test is only for planktonic bacteria while confocal can show live/dead percentage of biofilm. A result of the percentage of live/dead bacteria would be more promising showing the killing effect on biofilm.

A 2)

Thank you for the comment and valuable input. We agree with your comment and now we have carried out quantitative analyses using ImageJ software and the results were added into Figure 6.

Q 3)

Why L929 cells were used?

A 3)

L959 cells, which were murine fibroblasts have been considered in this study as they were the most frequently considered cells for biocompatibility in terms of the cytotoxicity endpoint. In fact, they have been recommended by the International Standards, ISO 10993-5 which outlines methods of considering cytotoxicity in biological materials. We have now included these details in the manuscript. (text colored red in line 130~132)

Q 4)

Why only two sites were measured on each sample for microhardness test? How did the authors determine this number?

A 4)

We have considered two sites for each sample in accordance to other previous studies. Still, we have also measured three samples of experimental group, and two sites were measured, which would provide total of 6 sites measured within the group. This has bene stated. (Text colored red in line 108~109)

Reviewer 2 Report

This MS describes the impact of polymerized PMMA-Phytoncide on C. albicans especially dental microbes. Although this preliminary report showed the antifungal activity of the prepared specimens, further clarifications are necessary before considering this MS for publication. Please try to clarify the following queries. 

Major comments 

The information about polymerized samples conversion into specimens must be included. Formation of phytoncide must be proved by analytical techniques like FTIR.

In all the experiments Phytoncide must be tested as control at equal concentration with the specimens and the datas need to be presented.

Only 1ml of fungal broth with specimen will dry at 37 C. In that case how the fungi cells stay alive? And author use sonication to separate the adhered cells which might disrupt the cells and could show negative results. Author should explain it in detail.

Author must explain how good fit the bacterial viability kit for fungi cells is. Although fungi are a microbe, but its physiological composition is entirely different from bacteria.

From where the control PMMA specimen got color? As far as the elsewhere available data PMMA discs are either white or yellow in color.

How could the cell viability be represented more than 100%?

Author should submit a clearly, magnified fluorescence microscopy images. The presented data provide nothing. How sure the author claimed that the tested saliva containing only C. albicans but not other microbes?

Topographical structure to be presented by advanced microscopy techniques like SEM, AFM.
Author must include discussion about the chemistry behind specimen formation of PMMA with phytoncide and how does it help to improve the physical properties of the specimen.

Since pH is an important factor in oral cavity author must consider testing their specimen’s activity at different physiological pH.

minor comments
In many places the intended meaning is wrong. Need a careful proof-reading by a scientific writer.
Typo errors through out the MS need to be corrected.

I would recommend a MAJOR REVISION for this MS.

Author Response

Reviewer #2

Q 1)

The information about polymerized samples conversion into specimens must be included. Formation of phytoncide must be proved by analytical techniques like FTIR.

A 1)

Thank you for your comments and valuable suggestions. As suggested, FTIR was now performed to evaluate the conversion. In the revised manuscript, detailed experimental methods and results were included as well as additional figure (Fig. 1).

Q2)

In all the experiments Phytoncide must be tested as control at equal concentration with the specimens and the datas need to be presented.

A2)

Thank you for your comments. We considered your comments and understood as the additional experiments with Phytoncide by itself (not incorporating into PMMA) at concentrations same as the Phytoncide-PMMA composite. Still, we believe that the effect is from the Phytoncide extracted from PMMA or located on the surface of PMMA. Hence, it would be difficult to quantify effective concentration.

Q3)

Only 1ml of fungal broth with specimen will dry at 37 C. In that case how the fungi cells stay alive?

A3)

Thank you for your consideration. Following the methodology in the previous study, similar aliquot proportion was adapted and moisture and temperature-controlled incubation unit was used to faciliate candida growth. Also, this was in the condition of more than 95% humidity. The details are now included in the manuscript (text colored red in line 153~154)

Q4)

And author use sonication to separate the adhered cells which might disrupt the cells and could show negative results. Author should explain it in detail.

A4)

Thank you for your comments. The method of using the sonication to separate adhered bacteria is now common practice of studying microbiological effect of biomaterials, and as we have considered live/dead images which are added as new part of Figure 6, it was evident that the disruption was minimal or none. The details are now included in methods (text colored red in line 155~157).

Q5)

Author must explain how good fit the bacterial viability kit for fungi cells is. Although fungi are a microbe, but its physiological composition is entirely different from bacteria.

A5)

Sorry for the confusion. We used the LIVE/DEAD® FungaLight™ Yeast Viability Kit, which enables researchers to easily, reliably and quantitatively distinguish live and dead yeast. The explanations are now included in the manuscript (text colored red in line 162~164).

Q6)

From where the control PMMA specimen got color? As far as the elsewhere available data PMMA discs are either white or yellow in color.

A6)

Sorry for another confusion. In this study, we used commercially available PMMA that is used as denture based materials in dentistry. The color used as control group is the color of polymer itself of commercially available Probase products. Colors may vary from other products.

Q7)

How could the cell viability be represented more than 100%?

A7)

The experimental group was higher than 100% because the control group was based on 100% when calculating cell viability (%). However, there is no statistical difference. Hence, there was no cell number difference between control and test samples in case where test samples showed 100% cell viability. The discussion has been added to provide explanations for such high viability (Line 289~305)

Q8)

Author should submit a clearly, magnified fluorescence microscopy images. The presented data provide nothing. How sure the author claimed that the tested saliva containing only C. albicans but not other microbes?

A8)

This would be another confusion caused by our manuscript and we do apologize for this. Our saliva consists of many other bacteria/fungi other than C. Albicans. The test on oral biofilm was performed to see if the material under research would be effective on biofilm. Hence as the title suggests, the experiment was aimed at multispecies oral biofilm for that part. The Figure 6 demonstrates oral biofilm thickness which would be composed of multispecies. The technique has been adapted from previous studies. The images have been updated and the details of this is provide in the text (line 315~321)

Q9)

Topographical structure to be presented by advanced microscopy techniques like SEM, AFM.

A9)

Thank you for your suggestion. We considered SEM images as below. There were no difference in topographical features. We have now included this as part of Figure 3.

Q10)

Author must include discussion about the chemistry behind specimen formation of PMMA with phytoncide and how does it help to improve the physical properties of the specimen.

A10)

Thank you for your valuable comments. We have now considered chemistry between PMMA and phytoncide with the results obtained from FT-IR (Figure 1). The results indicated the successful conjugation of the phytoncide with the polymer. Our results did not indicate improvement of physical properties. In fact, there has been decrease in flexural strength as amount of phytoncide has been increased. It has been well known from the previous study that polymerization may have been affected by conjugation between the added component and PMMA. This has been elaborated in Disucssion.

Q11)

Since pH is an important factor in oral cavity author must consider testing their specimen’s activity at different physiological pH.

A11)

Thank you for the suggestion. The present study design was a basic study to evaluate usefulness of the combination of phytoncide with commercially available PMMA. As appended in the discussion, within the standing limitations of the oral environment simulation and an in vitro model, the results are recommended to be interpreted in accordance. Further studies closely replicating the oral environmental variation such as temperature and pH are necessary for thorough understanding of phytoncide application.

Reviewer 3 Report

The research results presented in this manuscript concern mainly the biological/health aspect. In my understanding the material aspect concerns the denture resin in which is incorporated a natural substance with few concentrations. There are some results about the mechanical properties and few other physical caracteristics but the major results do not concern physics. I am not convinced that this manuscript is appropriate to this journal even though the authors have already published an article in it (materials). Besides, I found that the names of the used instruments are given and repeated without bringing a significant and useful information for the reader. Whatever is the decision of the editor about this manuscript, here are some comments page by page:

Page 1, abstract: L15. "their poor antifungal effects remain limited". This sentence is not clear. If their antifungal effects are poor and remain poor then the term limited is not necessary.

L40. as PMMA are --> as PMMA is.

L43-44. Hard to understand the meaning, english style to reconsider.

Page 2.

L60. Recently. repeated twice.

L61. Studying in containing. What is the meaning ? hard to understand.

L69-71. Hard to understand. To reconsider.

What is a null hypothesis ?

Page 3.

L93. Physical mechanical properties --> mechanical properties.

L104. mm^3. Use superscripts.

L107. What is the meaning of h2? Do you mean h scarred? You can put the expression in a separate line and number the equation.

Page 4. Names of instruments repeated.

Page 5. In the figures what is the meaning of a) b) and so on? Can comment the error bars?

Page 6. The size of figure 3 A) does not allow the reader to see the details. Can you comment the error bars on figure 3 B) ? the symbol a) ?

Page 7. Fig 4 A not of enough quality and size.

The last sentence in the captions of figs 4 and 5 not clear and must be completed.

Author Response

Reviewer #3

Q 1)

Page 1, abstract: L15. "their poor antifungal effects remain limited". This sentence is not clear. If their antifungal effects are poor and remain poor then the term limited is not necessary.

A 1)

Thank you for your careful considerations. The abstract has been modified to reflect above comments.

Q2)

L40. as PMMA are --> as PMMA is.

A2)

Thank you for your detailed review and sorry for such a mistake. We have corrected the grammar mistakes.

Q3)

L43-44. Hard to understand the meaning, english style to reconsider.

A3)

Introduction has been generally reviewed in terms of English style.

Q4)

Page 2.L60. Recently. repeated twice.

A4)

Sorry for such a mistake. The final checks of manuscripts were carried out by all of authors to ensure there are no misspelling etc.

Q5)

L61. Studying in containing. What is the meaning ? hard to understand.

A5)

Sorry for the confusion. Entire sentence has been reviewed and modified.

Q6)

L69-71. Hard to understand. To reconsider. What is a null hypothesis ?

A6)

The sentence has been modified and the null hypothesis are now included.

Q7)

Page 3.L93. Physical mechanical properties --> mechanical properties.

A7)

Thank you. I modified this content according to your suggestion.

Q8)

L104. mm^3. Use superscripts.

A8)

We have reworded the texts(colored red).

Q9)

L107. What is the meaning of h2? Do you mean h scarred? You can put the expression in a separate line and number the equation.

A9)

Thank you for your comments and we modified according to your suggestion.

Q10)

Page 4. Names of instruments repeated.

A10)

We have reviewed instruments names and repeat has been deleted.

Q11)

Page 5. In the figures what is the meaning of a) b) and so on? Can comment the error bars?

A11)

The error bars in Figure 2A show large overlap indicating the reduced probability of statistically significant differences between the groups. In contrast, Figure 2B, the non-overlapping error bars depict a higher probability of statistically significant differences between the groups. The presence of statistically significance (P<0.05) has been indicated with lowercase letter, where different letters indicate the presence of statistical significant differences between the groups. For example, ‘a’s above error bars for Figure 1A means that there is no statistical difference between all groups, while in Figure 1B, ‘a’ over 1.25%PT and ‘b’ over 2.5%PT means that there were significant difference between two. This has been emphasized in figure legends.

Q12)

Page 6. The size of figure 3 A) does not allow the reader to see the details. Can you comment the error bars on figure 3 B) ? the symbol a) ?

A12)

Thank you for your comments. Figure is now modified to reflect the comment. Also, the sentence structure has been formatted to improve clarity in reading.

Also figure legend has been modified to reflect above explanations.

Q13)

Page 7. Figure 4A not of enough quality and size.

A13)

We modified the figure size and quality has been enhanced.

Q14)

The last sentence in the captions of figs 4 and 5 not clear and must be completed.

A14)

We have completed the figure legends.

Round 2

Reviewer 1 Report

The authors have improved the paper with additional data and manuscript modifications. The paper now can be considered for publication. 

Author Response

Thank you for your kind comments, and thank you for your review.

Reviewer 2 Report

Hence, the revised version of the MS clearly satisfied the raised queries and included the necessary explanations, it could be now considered for next level. 
I would like to recommend the present form of the MS for publication. 

Author Response

Thank you for your comnment and kind review.

Reviewer 3 Report

Thanks for the authors for the improvement of the manuscript with the more correct phrasing and the explanations given as answers to the comments contained in the first report. There are still some minor spelling/typographical errors which need to be corrected. The originality of the manuscript is related to the study of the characteristics (chemical, physical, biological) of dental PMMA containing up to 5% wt phytoncide. Here are some detailed comments:

L19. Define MTT assay (colorometric system).

L39. "is an crucial" --> is a crucial

L69-70. PMMA that would improve provide antifungal effect. Choose one verb "improve" or "proivde".

L95, Table 1. Indicate what PT stands for when writing  1.25% PT for instance?

L113 & L 118. In equation (1), the authors use b for width and h for height. Are these lengths connected to those given in L 113? what are their values? is b equal to 3.3 mm or 10 mm? same thing for h. You can use in L113 for instance Lxlxh, where L is the length, l the width and h the height if this is correct.

L204. MMA to be defined

L207. Figure 1. Please put the numerical values for the y-axis (transmittance). There is also some thing which seems unusual. The wavenumbers are decreasing for left to right in Figure 1 when one expect the opposite.

L211. Physical mechanical properties. May be Mechanical properties is enough

L221-222. Still the meaning of lower-case letters a, b and c is not clearly explained especially the difference between a, b and c.

L253. Figure 5B --> Figure 6B

L254. As for lines 221-222, the authors use a, b and c letters. In addition, they have used "bc" for the case 3.75%PT. What does it mean?

Figure 6B. The y-axis label (biofilm thickness units. please replace um by microns or micro-meters).

L296. which has been a well-established method-->which is a well-established method

L301. group--> groups

L302. compare--> "in comparison to" or "as compared to"

L303. It has been well known-->It is well known

L308. that there have been a successful conjugation--> that there is a successful conjugation

Author Response

Q 1)

L19. Define MTT assay (colorometric system).

A 1)

We agree with your comment and then we add information in MTT assay

(text colored red in Line 19~20).

Q 2)

L39. "is an crucial" --> is a crucial

A 2)

Thank you for your comments. We revised the grammar mistakes (colored red).

Q 3)

L69-70. PMMA that would improve provide antifungal effect. Choose one verb "improve" or "proivde".

A 3)

Thank you for your careful consideration. We determined to choose on verb “improve”.

Q 4)

L95, Table 1. Indicate what PT stands for when writing 1.25% PT for instance?

A 4)

Sorry for the confusion. We indicated PT(colored red) in Table 1.

Q 5)

L113 & L 118. In equation (1), the authors use b for width and h for height. Are these lengths connected to those given in L 113? what are their values? is b equal to 3.3 mm or 10 mm? same thing for h. You can use in L113 for instance Lxlxh, where L is the length, l the width and h the height if this is correct.

A 5)

Sorry for such a mistake. We revised about specimen length, width, height according to ISO 20795-1. Then, these lengths not connected to those given in L113. l is the distance between the supports.

Q 6)

L204. MMA to be defined

A 6)

Thank you for your detailed review and we add sentence that is defined MMA

(text colored red in Line 205~206 ).

Q 7)

L207. Figure 1. Please put the numerical values for the y-axis (transmittance). There is also something which seems unusual. The wavenumbers are decreasing for left to right in Figure 1 when one expect the opposite.

A 7)

Thank you for your valuable comment. We put the numerical values for the y-axis.

Q 8)

L211. Physical mechanical properties. May be Mechanical properties is enough

A 8)

Thank you. I modified this content according to your suggestion.

Q 9)

L221-222. Still the meaning of lower-case letters a, b and c is not clearly explained especially the difference between a, b and c.

A 9)

Same lowercase alphabetical letters above the bar graph indicate there are no significant differences between the groups (e.g. there are no difference between group with ‘a’ and another group with ‘a’) (P > 0.05).

Difference in lowercase alphabetical letters above the bar graph indicate significant differences between the groups (e.g. there are difference between group with ‘a’ and group with ‘b’.) (P < 0.05).

In other words, for Microhardness, there were no difference between all of groups, and hence all labelled ‘a’.

For Flexural strength, there were no difference between control and 1.25% PT and hence both with ‘a’. However, there were difference between 1.25% PT and 2.5% PT and therefore one has ‘a’ and another has ‘b’ labelled.

Q 10)

L253. Figure 5B --> Figure 6B

A 10)

Sorry for the mistake. We modified this content.

Q 11)

L254. As for lines 221-222, the authors use a, b and c letters. In addition, they have used "bc" for the case 3.75%PT. What does it mean?

A 11)

Same lowercase alphabetical letters above the bar graph indicate there are no significant differences between the groups (e.g. there are no difference between group with ‘a’ and another group with ‘a’) (P > 0.05).

Difference in lowercase alphabetical letters above the bar graph indicate significant differences between the groups (e.g. there are difference between group with ‘a’ and group with ‘b’.) (P < 0.05).

There were no differences between 2.5% PT and 3.75% PT, and hence both ‘b’.

There were no differences between 3.75% PT and 5%PT, and hence both ‘c’.

Still there were difference between 2.5% and 5%PT and hence one is labeled b and another c.

Q 12)

Figure 6B. The y-axis label (biofilm thickness units. please replace um by microns or micro-meters).

A 12)

Thank you for your valuable comment. We modified biofilm thickness units to microns.

Q 13)

L296. which has been a well-established method-->which is a well-established method

A 13)

Thank you. We modified this content according to your suggestion.

Q 14)

L301. group--> groups

A 14)

Thank you for your consideration. We modified this content.

Q 15)

L302. compare--> "in comparison to" or "as compared to"

A 15)

Thank you for the suggestion. This sentence is now modified to reflect the comment.

Q 16)

L303. It has been well known-->It is well known

A 16)

We modified this sentence according to your comment.

Q 17)

L308. that there have been a successful conjugation--> that there is a successful conjugation

A 17)

Thank you for this comment. We have reworded the sentence.